# Development of Tumor Cell-Based Vaccine with IL-12 Gene Electrotransfer as Adjuvant

**DOI:** 10.3390/vaccines8010111

**Published:** 2020-03-02

**Authors:** Tinkara Remic, Gregor Sersa, Katja Ursic, Maja Cemazar, Urska Kamensek

**Affiliations:** 1Department of Experimental Oncology, Institute of Oncology Ljubljana, Zaloska cesta 2, SI-1000 Ljubljana, Slovenia; tremic@onko-i.si (T.R.); gsersa@onko-i.si (G.S.); kursic@onko-i.si (K.U.); mcemazar@onko-i.si (M.C.); 2Faculty of Medicine, University of Ljubljana, Vrazov Trg 2, SI-1000 Ljubljana, Slovenia; 3Faculty of Health Sciences, University of Ljubljana, Zdravstvena pot 5, SI-1000 Ljubljana, Slovenia; 4Faculty of Health Sciences, University of Primorska, Polje 42, SI-6310 Izola, Slovenia

**Keywords:** tumor cell-based vaccine, IL-12, gene electrotransfer, radiotherapy, B16-F10, CT26

## Abstract

Tumor cell-based vaccines use tumor cells as a source of tumor-associated antigens. In our study, we aimed to develop and test a tumor vaccine composed of tumor cells killed by irradiation combined with in vivo interleukin-12 gene electrotransfer as an adjuvant. Vaccination was performed in the skin of B16-F10 malignant melanoma or CT26 colorectal carcinoma tumor-bearing mice, distant from the tumor site and combined with concurrent tumor irradiation. Vaccination was also performed before tumor inoculation in both tumor models and tumor outgrowth was followed. The antitumor efficacy of vaccination in combination with tumor irradiation or preventative vaccination varied between the tumor models. A synergistic effect between vaccination and irradiation was observed in the B16-F10, but not in the CT26 tumor model. In contrast, up to 56% of mice were protected from tumor outgrowth in the CT26 tumor model and none were protected in the B16-F10 tumor model. The results suggest a greater contribution of the therapeutic vaccination to tumor irradiation in a less immunogenic B16-F10 tumor model, in contrast to preventative vaccination, which has shown greater efficacy in a more immunogenic CT26 tumor model. Upon further optimization of the vaccination and irradiation regimen, our vaccine could present an alternative tumor cell-based vaccine.

## 1. Introduction

The induction of effective antitumor immunity begins with the presentation of tumor-associated antigens (TAA) to antigen-presenting cells, such as dendritic cells (DC), through which the immune system can mount a specific antitumor response [1]. However, during oncogenesis, tumor cells evolve to avoid the immune system [2]. Various immunotherapy approaches aim at restoring antitumor immunity. Two such approaches that take advantage of patients’ own TAA are modified whole tumor cell vaccines [3,4,5,6,7] and vaccines prepared from tumor cell lysates [3,8,9,10,11,12,13,14]. 

Whole tumor cell vaccines are composed of patient-derived tumor cells which are ex vivo genetically modified to secrete immunological adjuvants and then inactivated by irradiation [1,3,4,5,6,7]. These include GVAX and Algenpantucel-L, currently under clinical evaluation, composed of tumor cells transduced with granulocyte-macrophage colony stimulating factor and α-1,3-galactosyltransferase, respectively [6,7]. Alternatively, tumor cells can be killed ex vivo to prepare tumor cell lysates using several methods, including ionizing irradiation, freeze-thaw cycles, and high hydrostatic pressure [8,9,10]. Lysates, containing TAA, can then be used directly as vaccines or be pulsed into patient-derived DC to prepare DC vaccines [3,8,9,10,11,12,13,14]. Examples of such vaccines include Provenge, an Food and Drug Administration approved DC-based vaccine [12,13], and a melanoma tumor lysate vaccine prepared by a cytotoxic peptide [14].

As with standard vaccination using defined TAA, vaccination with tumor-derived TAA greatly benefits from immunological adjuvants that are usually delivered by viral vectors [3,4,5,6,7,12,13]. But genes encoding immunological adjuvants can also be delivered by non-viral methods. One of the most utilized non-viral methods for gene delivery is gene electrotransfer (GET) that was proven promising for in vivo localized delivery of pro-inflammatory cytokine interleukin-12 (IL-12) [15,16]. IL-12 GET was tested for localized intratumoral or peritumoral delivery or as an adjuvant to vaccines administered intradermally or intramuscularly in numerous preclinical studies [17,18,19,20,21,22,23,24,25] and is currently under clinical evaluation [26].

Routine cancer therapies are usually multimodal. Immunotherapies are often combined with tumor ablation therapies, such as radiotherapy. Aside from the obvious tumor ablation effects, radiotherapy also has immunological effects. It has the ability to shift the tumor microenvironment from immunosuppressive to immunoreactive through several events, including the enhancement of antigen presentation by major histocompatibility class I molecules (MHC-I) [27,28,29], transient normalization of tumor vasculature that enables a greater immune cell infiltration, and induction of immunogenic cell death [30,31,32].

Our previous approach to vaccination was in situ vaccination using electrochemotherapy to release the TAA, which we then boosted with intradermal IL-12 GET around the tumor, i.e., peritumorally [17,18,19,20]. In this study, we aimed to simulate this phenomenon distant from the tumor for potential treatment of deep-seated tumors that cannot be reached by electrochemotherapy. In the clinical setting, such a vaccine could be prepared from patient-derived tumor cells acquired during surgery and introduced back to the patient concomitantly to conventional cancer treatment by simple intradermal vaccination, boosting the patient’s immune response against the TAA in the vaccine and released from the treated tumors. 

Hence, the aim of this study is to develop a tumor cell-based vaccine with IL-12 GET as an immunological adjuvant. To prove the concept, we tested different vaccination protocols using syngeneic tumor models. Our vaccine was prepared from tumor cells killed by irradiation we injected in the skin distant from the tumor site and combined with IL-12 GET. Vaccination was tested in combination with tumor irradiation and in the preventative setting, in two immunologically different tumor models [33,34]. The results confirmed the synergism of concomitant vaccination and radiotherapy in the less immunogenic malignant melanoma B16-F10 model, but not in the more immunogenic colon carcinoma CT26 model. Additionally, the results showed the preventative effect of vaccination in the CT26 tumor model, but not in the B16-F10 tumor model.

## 2. Materials and Methods

### 2.1. Cell Lines

Murine melanoma B16-F10 (ATCC, Manassas, VA, USA) and colon carcinoma CT26 (ATCC, Manassas, VA, USA) cell lines were cultured in advanced modified Eagles medium (A-MEM; Thermo Fisher Scientific, Waltham, MA, USA) and advanced Roswell Park Memorial Institute medium (A-RPMI; Thermo Fisher Scientific, Waltham, MA, USA), respectively. Both culture media were supplemented with 5% fetal bovine serum (FBS; Thermo Fisher Scientific, Waltham, MA, USA), 10 mM L-glutamine (Thermo Fisher Scientific, Waltham, MA, USA), 50 mg/mL gentamicin (Krka, Novo Mesto, Slovenia), and 100 U/mL penicillin (Sandoz International GmbH, Holzkirchen, Germany). Cells were grown at 37 °C in a humidified atmosphere containing 5% CO_2_ and were used within ten passages.

### 2.2. Plasmid DNA

Plasmid DNA pORF-mIL-12-ORT encoding murine IL-12 without an antibiotic resistance gene [35] was isolated using the EndoFree Plasmid Mega Kit (Qiagen, Hilden, Germany) and dissolved in endotoxin-free water to a final concentration of 0.625 mg/mL. Plasmid purity and the concentration were determined using the Epoch Microplate Spectrophotometer, Take3™ Micro-Volume Plate (BioTek, Bad Friedrichshall, Germany). The plasmid identity was confirmed by restriction enzyme analysis on an electrophoretic gel.

### 2.3. Animals

In vivo experiments were performed in 6–8 week old female C57BL/6 and BALB/c mice (Charles River, Sant’Angelo Lodigiano, Italy). Mice were maintained in animal colony in a 12 h light/dark cycle under specific pathogen-free conditions at constant room temperature and humidity. Food and water were provided ad libitum. Before the experiments, animals were subjected to a quarantine and an adaptation period of 2 weeks. All experimental procedures were performed in accordance with the EU directive (2010/63/EU) and with the guidelines of the Ministry of Agriculture, Forestry, and Food of the Republic of Slovenia (permission no. U34401–1/2015/38). B16-F10 and CT26 tumors were induced with a subcutaneous injection of 1 × 10^6^ viable B16-F10 or 0.5 × 10^6^ viable CT26 tumor cells into the lower backs of respective syngeneic C57BL/6 and BALB/c mice. When the tumors reached a specific volume (35–40 mm^3^), the mice were randomly divided into different treatment groups and treated according to a specific experimental protocol. During vaccination and GET, the mice were anesthetized with isoflurane anesthesia. The mice were sacrificed using a CO_2_ chamber or via cervical dislocation.

### 2.4. Irradiation

Cells were irradiated at a dose rate of 1.728 Gy/min using the Darpac 3300 X ray unit (Gulmay Medical Ltd., Byfleet, Surrey, UK) operating at 200 kV and 9.2 mA with a 0.55 mm Cu and 1.8 mm Al filtration. Animals were irradiated under the same conditions while being restrained in special lead tubes with fixed apertures.

### 2.5. Vaccination

#### 2.5.1. Pilot B16-F10 Vaccination

For vaccine preparation, B16-F10 cells were irradiated with an empirically chosen lethal single dose of 25 Gy (Figure 1a). After 10 days, the medium containing non-viable tumor cells (NTC) was collected and centrifuged for 5 min at 470 g (Heraeus Multifuge 1S-R; Thermo Fisher Scientific, Waltham, MA, USA). The vaccine was prepared to a final concentration of 0.5 mg of proteins per a unit of vaccine as measured using the Pierce™ bicinchoninic acid (BCA) Protein Assay Kit (Thermo Fisher Scientific, Waltham, MA, USA).

On the day of the treatment, C57BL/6 mice were randomly divided into treatment groups (n = 6 mice per group). One unit of the pilot vaccine or the mock vaccine was injected subcutaneously distant from the tumor in the upper back of mice. IL-12 GET was performed by intradermal injections, 4 × 20 µL of IL-12 plasmid (50 µg) at four locations forming an approximately 1 cm^2^ area surrounding the vaccine injection site (Figure 1b). A contact multielectrode array (MEA, Iskra Medical, Ljubljana, Slovenia) was positioned to encompass the injected IL-12 plasmid between the electrodes and 12 electric pulses (170 V/cm, 2.82 Hz, 150 ms) were applied [17]. Tumors were concomitantly irradiated with 10 Gy. 

#### 2.5.2. Adjusted B16-F10 and CT26 Vaccine Preparation and Therapeutic Vaccination

Because of the results of the pilot study, which showed the remaining viable cells at the NTC site and a higher amount of immune cells at the site of IL-12 GET than at the NTC site, adjustments were made to the above-described vaccine preparation procedure (Figure 1a). B16-F10 and CT26 cells were irradiated first with three fractions of 5 Gy. Two days after the final fraction, a dose of 30 Gy was applied that killed the B16-F10 cells, but not all the CT26 cells. A second dose of 30 Gy was required to kill the remaining CT26 cells 5 days after the first 30 Gy dose. Five days after final irradiation, both B16-F10 and CT26 NTC were collected and centrifuged for 5 min at 470 g. To prepare a vaccine with a higher concentration, the collected medium without NTC was concentrated at 30 °C, using the V-AQ program of the Concentrator plus (Eppendorf, Hamburg, Germany). The vaccine was prepared to a final concentration of 0.5 mg/U or 1 mg/U as measured by the Pierce™ BCA protein Assay Kit and stored at −80 °C.

On the day of the treatment, C57BL/6 and BALB/c mice were randomly divided into treatment groups (n = 6–12 mice per group). The IL-12 plasmid (50 µg) was dissolved in 100 µL of NTC (0.5 mg/U or 1 mg/U). The mixture represented 1 unit of the B16-F10 or CT26 vaccine containing 0.5 mg or 1 mg of protein. One unit of the vaccine or the mock vaccine was injected subcutaneously distant from the tumor in the upper back of the mice. To perform GET of the IL-12 plasmid contained within the vaccine, a contact MEA, including a central pin, was positioned to encompass the injected vaccine, and 24 electric pulses (170 V/cm, 5.64 Hz, 150 ms) were applied (Figure 1b). Tumors were concomitantly irradiated with 15 Gy. 

#### 2.5.3. B16-F10 and CT26 Preventative Vaccination

The ability of the B16-F10 and CT26 vaccine prepared using the adjusted vaccine preparation protocol to prevent tumor outgrowth was tested in B16-F10 and CT26 tumor models. C57BL/6 and BALB/c mice were randomly divided into treatment groups (n = 8–9 mice per group). Seven or 14 days before tumor inoculation, vaccination was performed as described for the adjusted therapeutic vaccination protocol. After tumor inoculation, mice were followed for any signs of tumor growth. If the tumors grew, the tumor growth was followed. 

### 2.6. Treatment Effectiveness Assessment

Tumor volumes were calculated using the formula: $V=\frac{a\cdot b\cdot c\cdot \pi }{6}$; where a, b, and c correspond to the three orthogonal diameters of the tumor, which were measured every second day using a Vernier Caliper [36]. Treatment started when tumor volume reached 35–40 mm^3^. Tumor growth delay was calculated using the formula: $\mathrm{GD}={\bar{\mathrm{DT}}}_{\mathrm{tr}}-{\bar{\mathrm{DT}}}_{\mathrm{untr}}$; where DT_tr_ and DT_untr_ correspond to the average doubling times of the treated or untreated tumors, respectively. A complete response was defined as the absence of a detectable tumor for 100 days. The general well-being of mice was monitored by their weight, their ease of movement, and their behavior. The vaccination site was observed for any signs of reactions to the vaccination. The humane endpoint was a tumor size of 250 mm^3^, because of the nature of tumors (prone to exulceration after tumor irradiation).

### 2.7. Histology

The skin at the vaccination site and tumors were harvested 3 and 6 days following the start of the treatment. Samples were left overnight in a formalin-free zinc fixative (BD Biosciences, San Jose, CA, USA) and then transferred into 70% ethanol. Skin and tumor samples were embedded in paraffin and 5 or 2 µm thick sections were cut, respectively. Samples were stained with hematoxylin and eosin (H&E) and observed at 40 × magnification using a BX-51 microscope (Olympus, Düsseldorf, Germany). 

Immunohistochemical (IHC) staining was performed using the EXPOSE Rabbit HRP/AEC kit (ab64261; Abcam, Cambridge, UK) and Rabbit Specific HRP/AEC IHC Detection Kit-Micro-polymer (ab236468; Abcam, Cambridge, UK) following the manufacturer protocol. The primary Anti-Granzyme B antibody (ab4059; Abcam, Cambridge, UK) at 1:1000 dilution and the primary Foxp3 Antibody (5H10L18; Thermo Fisher Scientific, Waltham, MA, USA) at 1:1200 dilution were used. Samples were observed at 40 × objective magnification using the BX-51 microscope. For analysis, 10 images per sample were taken using a DP72 CCD camera connected to the microscope. The average number of Granzyme B positive (GrB+) and FoxP3 positive (FoxP3+) cells in a visual field was assessed in a blind fashion by three examiners [21,36].

### 2.8. Statistical Analysis

GraphPad Prism version 8.1.2. (GraphPad Software, San Diego, CA, USA) was the software used for data analysis and graphical presentations. The Shapiro–Wilk test was performed to test the data normality. Normally distributed data were presented as the mean ± standard error (SE), while non-normally distributed data were presented as the median with data range (min, max). Normally distributed data with equal variance were analyzed with a one-way ANOVA, followed by Tukey’s test for multiple comparisons. Normally distributed data with unequal variance were analyzed with a Browne–Forysthe and Welch’s ANOVA, followed by Dunnet’s T3 test for multiple comparisons. Non-normally distributed data were analyzed with a Kruskal–Wallis ANOVA on Ranks, followed by Dunn’s test for multiple comparisons. 

A two-tailed unpaired t-test was performed to compare the radiosensitivity of the B16-F10 and CT26 tumor models. The combination index (CI) was used to determine whether the combination of two treatments with an independent mechanism of action had a synergistic (CI < 0) effect using a formula that takes into account the statistical variability inherent in biological systems [37]. The preventative efficacy of vaccination was analyzed with a Kaplan–Meier survival analysis followed by a logrank test for multiple curve comparison. Tumor occurrence was defined as an event and a complete response as a censored event. A two-tailed unpaired t-test for normally distributed data (or a Mann–Whitney U-test for non-normally distributed data) was performed to compare the numbers of GrB+ cells on different days within a treatment group. A significant difference between experimental groups was defined as *p* < 0.05.

## 3. Results

### 3.1. Pilot B16-F10 Vaccination

An in vivo study was performed in the B16-F10 tumor model to determine the efficacy of the pilot vaccination, i.e., NTC surrounded by IL-12 GET, distant from the locally irradiated tumor (10 Gy). The longest tumor growth delay was observed in mice receiving the therapeutic combination; however, the contribution of the pilot vaccine to tumor irradiation was not significant (Figure 2a,b, Appendix A). H&E stained histological sections of the vaccination sites showed a higher infiltration of immune cells to the site where IL-12 GET was performed than to the site of the NTC injection (Figure 3c). Additionally, in one case, viable tumor cells were observed at the NTC injection site (Figure 3b). 

### 3.2. Adjusted B16-F10 and CT26 Vaccination

The vaccination protocol was adjusted according to the result of the pilot study by administering a mixture of NTC and IL-12 plasmid as the adjusted vaccine. The therapeutic effect of the adjusted vaccine (VAC) in combination with tumor irradiation with 15 Gy (IR) was tested in two immunologically different tumor models: the B16-F10 and CT26 [33,34]. In the B16-F10 model, the adjusted vaccination significantly contributed to tumor irradiation. A significantly longer tumor growth delay was observed in the combined treatment groups receiving the vaccination with both vaccine doses and irradiation (VAC (0.5 mg) + IR, VAC (1 mg) + IR) (Figure 4, Appendix A). The growth delay was longer with the higher vaccination dose, though not statistically significant. Among the control groups, irradiation alone was the most effective and no significant contribution of IL-12 GET or NTC to tumor irradiation was observed. Synergism between the tumor irradiation and both vaccination doses was confirmed (CI < 0). 

In the CT26 tumor model however, vaccination did not significantly contribute to tumor irradiation. The only difference in tumor growth delay was between mice that did or did not receive tumor irradiation (Figure 5, Appendix A). In all the groups receiving irradiation, there were some complete responses (17–50%), while there were none in the B16-F10 tumor model. In the CT26 tumor model, tumor irradiation alone led to a 22.8 ± 2.7 days of tumor growth delay, which was significantly higher compared to the tumor growth delay after irradiation in the B16-F10 tumor model, where the delay was only 1 ± 0.6 day. After vaccination in both tumor models, a delayed type hypersensitivity-like reaction was observed at the vaccination site in most mice (Appendix A). 

### 3.3. B16-F10 and CT26 Preventative Vaccination

To confirm the systemic efficacy of the vaccination, both doses of the B16-F10 and CT26 vaccine, prepared using the adjusted vaccine preparation protocol, were tested in a preventative setting, 7 and 14 days before tumor inoculation. Interestingly, the B16-F10 vaccine did not prevent tumor outgrowth of B16-F10 tumors regardless of the time of administration and the dose (Figure 6a). On the other hand, the CT26 vaccine successfully prevented the outgrowth of CT26 tumors. The higher vaccine dose administered 14 days before tumor inoculation (VAC (1 mg) 14 days) was the most effective, preventing tumor outgrowth in 55.6% of mice (*p* < 0.05 against control and IL-12 GET 7 or 14 days; Figure 6b, Appendix A). The lower vaccine dose and higher NTC dose administered 14 days before tumor inoculation (VAC (0.5 mg) 14 days, NTC (1 mg) 14 days) also prevented tumor outgrowth in 33% and 25% of mice, respectively. Additionally, the administration of the higher vaccination dose 7 days before tumor inoculation (VAC (1 mg) 7 days) resulted in 22.2% of tumor free mice. Preventative vaccination did not affect the tumor growth dynamics once tumors occurred in either of the tumor models (Appendix A). After vaccination in both tumor models, delayed type hypersensitivity-like reaction was observed in most mice (Appendix A).

### 3.4. Histology

After adjusted vaccination, skin at the vaccination site and tumors were taken for histological analysis. The H&E stained sections of the skin showed immune cell infiltration to the site of vaccination in both tumor models. No viable cells were observed at the vaccination sites (Appendix A). 

Immunohistochemical staining of skin and tumor samples was performed to evaluate the presence of FoxP3+ and GrB+ cells. FoxP3+ cells were not detected in either the skin or tumor samples in either tumor model. In the B16-F10 tumor model, the number of GrB+ cells in skin samples on day 3 was significantly higher in the group receiving vaccination in combination with tumor irradiation (VAC + IR) compared to the control group (Figure 7). However, this was not significant compared to the group receiving only tumor irradiation (IR). Similarly, in tumor samples on day 3 the number of GrB+ cells was significantly higher in the combined treatment and irradiation groups (VAC + IR, IR) than in the control group (Figure 7). On day 6, in the control group the number of GrB+ cells in tumor samples, and interestingly also in skin samples, was significantly higher than on day 3. Representative images of each group in the B16-F10 tumor model are shown in Figure 7.

In the CT26 tumor model, in general, the number of GrB+ cells was much higher compared to the B16-F10 tumor model in both control skin and tumor samples. On day 3, the number of GrB+ cells in the combined treatment group (VAC + IR) was not higher in either the skin or tumor samples than in the control group (Figure 8). Yet, it was significantly higher in the group receiving only tumor irradiation (IR). However, on day 6, the number of GrB+ cells in the skin was significantly higher in the combined treatment group (VAC + IR) compared to the irradiation alone (IR) or the control group. The same was not true for tumor samples. Compared to day 3, the number of GrB+ cells in the combined treatment group (VAC + IR) on day 6 was significantly higher in skin samples, and lower in the tumor samples (Figure 8). Representative images of each group in the CT26 tumor model are shown in Figure 8. 

## 4. Discussion

Our study confirmed the feasibility of our tumor cell-based vaccine combining the irradiation-killed tumor cells and IL-12 GET as an immunological adjuvant. The results confirmed synergism between vaccination and tumor irradiation in the malignant melanoma B16-F10 model. However, the results also indicate that the effectiveness is tumor type-dependent, probably related to tumor immunogenicity. Specifically, the vaccine did not contribute to the effectiveness of irradiation in the more immunogenic colon carcinoma CT26 model, while its preventative effectiveness was evident in the CT26, but not in the less immunogenic B16-F10 tumor model.

In the first part of the experiments, we focused on the vaccine preparation protocol. In the pilot study, the dose of 25 Gy was chosen empirically by observing the cell viability after exposure to various radiation doses in vitro and choosing the dose where no viable cells remained. A unit of NTC was injected in the skin of the B16-F10 tumor-bearing mice distant from the tumor and IL-12 GET was performed around the injection site. The results showed a promising, yet insufficient, efficacy of the vaccine in combination with tumor irradiation. In H&E stained sections of the vaccination sites, we observed residual viable tumor cells and a higher immune cell infiltration to the IL-12 GET site than to the NTC injection site. 

To improve our vaccine for further experiments, we changed the irradiation regimen for vaccine preparation. Instead of 25 Gy, tumor cells were exposed to a final lethal dose of 30 Gy to ensure no viable tumor cells were present in the vaccine. Furthermore, to ensure immune cell infiltration coincided with the NTC injection site, NTC were mixed with the IL-12 plasmid, and electroporation was performed to encompass the vaccine mixture, thus enabling IL-12 GET. Additionally, to increase the immunogenicity of the vaccine, cells were exposed to 3 × 5 Gy before the lethal dose. Namely, several studies have reported an increase in the expression of immunomodulatory molecules because of the irradiation with doses as low as 2 Gy [30,32]. While at higher doses the immunomodulatory effects were reported to be counteracted via expression of exonuclease Trex1 [38]. Furthermore, elevated MHC-I molecule exposure was reported already after single doses of 4 Gy [27,28,29].

Both B16-F10 and CT26 vaccines prepared using the adjusted protocol were tested in B16-F10 and CT26 tumor-bearing mice. The adjusted B16-F10 vaccine was more effective than the pilot vaccine, confirming that the vaccine preparation was well adjusted. In H&E stained histological sections of vaccination sites, no viable cells were observed, and immune cell infiltration coincided with the vaccine mixture injection site in both tumor models. The successful induction of an immune response was also indicated by a delayed type hypersensitivity-like reaction at the vaccination sites, which is generally considered a positive treatment response predictor [39]. 

The therapeutic effects of the tumor cell-based vaccines, prepared using the adjusted protocol, were examined using two different vaccination doses. In the B16-F10 tumor model, a synergistic effect between vaccination with both doses and tumor irradiation was confirmed. This suggests that tumor irradiation may be required to elicit the effect of the vaccine in this tumor model. Namely, irradiation is known to induce an immunoreactive tumor microenvironment [30,40,41]. Therefore, tumor irradiation could have provided the necessary signals that enabled the shift of the immune response generated toward the TAA within the vaccine to the tumor site. 

Because of the promising results of the vaccination in the B16-F10 tumor model, we continued with testing the vaccine in the more immunogenic CT26 tumor model [33,34], with which we have previous laboratory experience. Unexpectedly, vaccination did not significantly contribute to tumor irradiation in the CT26 tumor model, regardless of the vaccination dose. On the other hand, the antitumor effect of tumor irradiation with 15 Gy alone was significantly higher in the CT26 tumor model compared with the B16-F10 tumor model. Therefore, the high efficacy of tumor irradiation might have concealed any contribution of vaccination in the CT26 tumor model. In the future, we are planning to use an equieffective dose of tumor irradiation to determine the contribution of vaccination to tumor irradiation. 

Furthermore, we tested the ability of the vaccine to elicit a systemic immune response by testing its efficacy in preventing tumor outgrowth in both tumor models. Surprisingly, we observed a promising preventative efficacy of vaccination in the CT26 tumor model and none in the B16-F10 tumor model. The reason is likely the difference in tumor immunogenicity. While CT26 is known as a more immunogenic tumor model with high MHC-1 expression [33,34], B16-F10 is considered less immunogenic, mainly because of the low MHC-I expression [33,34]. Therefore, we believe that, in the CT26 tumor model, the immune response primed against TAA in the vaccine recognizes the TAA on tumor cells and eliminates them, while, in the B16-F10 tumor model, the targets are hidden due to the low MHC-I expression. The latter was also reflected in the described synergistic effect radiation had on vaccination in B16-F10 tumor model. Namely, irradiation is known to heighten the MHC-I expression [27,28,29] thereby exposing the TAA. 

To further evaluate the local and systemic immune response to our vaccine, we assessed infiltrating immune cells at the vaccination site and in tumors by immunohistological staining for GrB+ and FoxP3+ cells. Tumor-infiltrating lymphocytes, such as GrB+ natural killer cells or effector T cells, are regarded as a positive prognostic marker in a clinical setting [42], while FoxP3+ regulatory T cells represent immune suppression, and are associated with an unfavorable outcome in a wide range of tumors [43]. In the B16-F10 tumor model, skin and tumor samples, stained for the presence of GrB, showed a higher median number of GrB+ cells on day 3 in the combined treatment group compared to the irradiated group. This trend coincided with the confirmed synergistic effect of tumor irradiation and vaccination on tumor growth.

Untreated CT26 tumors have been shown to have an inherently higher number of immune cells, compared to the B16-F10 tumor model [33], which was also confirmed in our study. Unlike in the B16-F10 tumor model, the number of GrB+ cells in CT26 tumor and skin samples on day 3 was higher after tumor irradiation alone, than after the main therapeutic combination. This may indicate that immune suppression has occurred in the group receiving the therapeutic combination. 

No FoxP3+ cells were detected in either tumor model, eliminating regulatory T cell-mediated immune suppression. Therefore, other possible mechanisms of immune suppression may have occurred in the CT26 tumor model, such as myeloid-derived suppressor cell-mediated or programmed cell death receptor mediated immune suppression [44,45]. It is possible the danger signals at the site of vaccination prevailed over the danger signals in tumors and the immune suppression was the result of damping down an overexcited immune system [46]. To yield a greater therapeutic effect of the vaccine, it may be necessary to lower the danger signals at the site of vaccination. Using alternative adjuvants such as granulocyte-macrophage colony-stimulating factor or IL-18 may enable this [47,48]. Or perhaps an adjuvant is redundant in this tumor model, as no benefits of adding IL-12 GET to NTC were observed. Another alternative would be to heighten the immunogenic danger signals at the tumor site, for instance by using a more immunogenic irradiation regimen or with checkpoint inhibitors [27,28,29,30,38,40,41].

Although in early stages of development, our vaccine combines the advantages of a tumor cell-based vaccine, i.e., targeting of multiple TAA [3,4,5,6,7,8,9,10,11,12,13,14,49,50,51,52,53], with a potentially inexpensive and time-saving vaccine preparation. By using GET to transfect the IL-12 plasmid, or any other adjuvant encoding plasmid, we bypass the ex vivo viral gene transfer used by whole cell and tumor lysate vaccines in development [3,4,5,6,7,49,54]. Namely, GET enables a highly effective gene transfer directly in vivo, which significantly shortens the time needed for the preparation of the autologous vaccine [3,4,5].

Apart from the mentioned advantages, our current vaccination protocol has some drawbacks. Although our therapeutic combination proved to be effective, synergistic even, in the B16-F10 tumor model, it did not lead to any complete responses in the combined treatment. However, it proved very successful in preventing tumor outgrowth in the CT26 tumor model. An optimized vaccination regimen may potentiate the antitumor efficacy of our vaccine. Namely, previous clinical and pre-clinical studies showed whole cell vaccinations and tumor lysate vaccines elicited a more potent antitumor response when they are repeated [4,50,51,52,53]. Likewise, further optimization of the irradiation regimens may facilitate the immune recognition of tumor cells by increasing the expression of danger signals and MHC-I presentation of TAA, as previously stated [27,28,29,30,31,32,38]. Furthermore, studies in various tumor models with different immune profiles such as colorectal carcinoma MC38, renal adenocarcinoma RENCA, or mammary adenocarcinomas TS/A and 4T1 tumor models, and patient-derived xenografts, may deepen our understanding of our vaccine’s mechanisms of action. 

## 5. Conclusions

As proof of principle, we showed that a single vaccination in combination with tumor irradiation can be effective in the B16-F10 tumor model. The results suggest a greater contribution of the vaccination to tumor irradiation in the less immunogenic tumor model, while, in a preventative setting, a greater contribution of the vaccination was indicated in the more immunogenic tumor model. Further studies are required to support the above statement. Upon successful establishment of the vaccination regimen as well as irradiation dosage and regimen, the vaccine, developed in this study, will present an alternative to tumor cell-based vaccines currently under clinical evaluation. 

## Figures and Tables

**Figure 1 vaccines-08-00111-f001:**
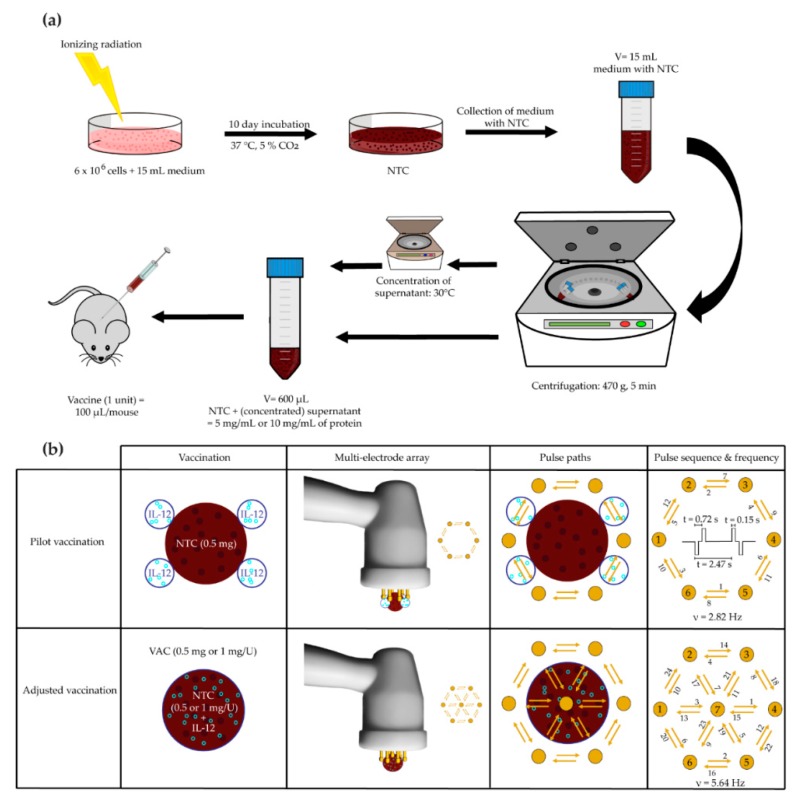
Overview of the tumor vaccine preparation procedure and application. (**a**) Tumor vaccine preparation. (**b**) Tumor vaccine application. NTC = non-viable tumor cells; IL-12 = interleukin-12; VAC = 1 unit of the vaccine (NTC and IL-12), prepared using the adjusted vaccine preparation protocol, including gene electrotransfer of IL-12.

**Figure 2 vaccines-08-00111-f002:**
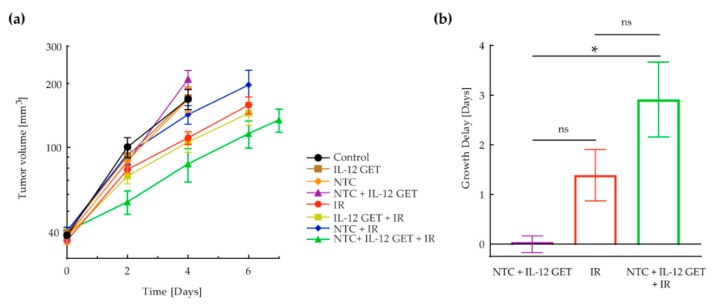
Pilot B16-F10 vaccination. (**a**) Tumor growth curve. Tumor volume in a logarithmic scale was plotted against time. (**b**) Tumor growth delay based on tumor doubling time. Eight treatment groups with six mice were included in the experiment. The control group received a mock vaccine. The IR group received tumor irradiation with 10 Gy. The NTC group received NTC. The IL-12 GET group received IL-12 GET. The NTC + IL-12 GET group received NTC and IL-12 GET. The IL-12 GET + IR and the NTC + IR groups received tumor irradiation and either IL-12 GET or NTC, respectively. The main therapeutic group was the NTC + IL-12 GET + IR group, which received tumor irradiation, NTC and IL-12 GET. * = *p* < 0.05; ns = not significant; NTC = 1 unit of non-viable B16-F10 tumor cells (0.5 mg); GET = gene electrotransfer; IR = tumor irradiation with 10 Gy.

**Figure 3 vaccines-08-00111-f003:**
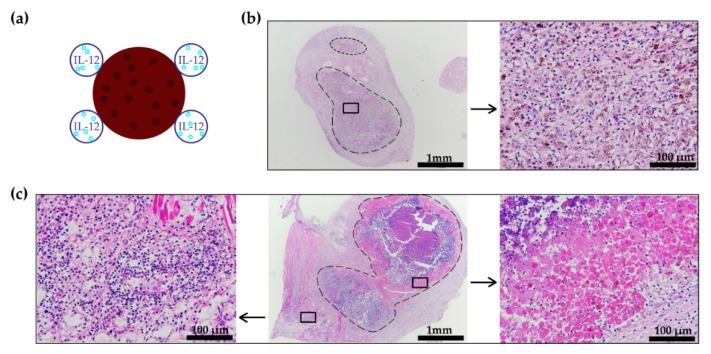
Pilot B16-F10 vaccination histology. (**a**) Schematic presentation of the vaccine application and IL-12 GET. (**b**,**c**) H&E stained histological section of the skin at the location of the pilot vaccine injection and IL-12 GET. The dashed outline marks the area of the injected pilot vaccine. The squares mark further magnified areas. (**b**) The magnified image shows viable cells at the location of the injected NTC. (**c**) The right magnified image indicates a lower immune cell infiltration to the location of the injected NTC than to the site where IL-12 GET was performed, as shown on the left magnified image. H&E = hematoxylin and eosin staining. GET = gene electrotransfer; NTC = 1 unit of non-viable B16-F10 tumor cells (0.5 mg).

**Figure 4 vaccines-08-00111-f004:**
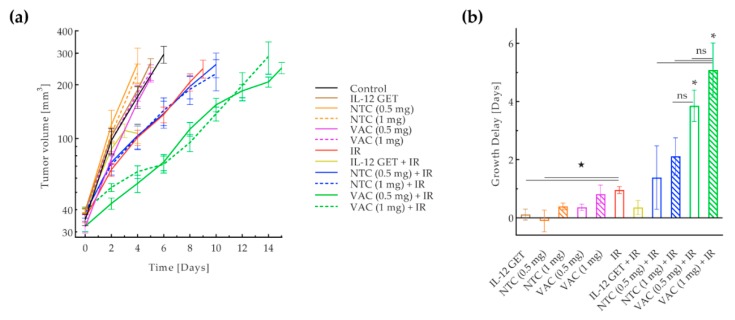
Adjusted B16-F10 vaccination. (**a**) Tumor growth curve. Tumor volume in a logarithmic scale plotted against time. (**b**) Tumor growth delay based on the tumor doubling time. Seven treatment groups with 6–12 mice were included in the experiments. The control group received a mock vaccine. The IL-12 GET group received IL-12 GET. The NTC (0.5 mg) and NTC (1 mg) groups received 0.5 mg or 1 mg of B16-F10 NTC, respectively. The VAC (0.5 mg) and VAC (1 mg) groups received 0.5 mg or 1 mg of the B16-F10 vaccine including IL-12 GET, respectively. The IR group received tumor irradiation with 15 Gy. The IL-12 GET + IR, NTC (0.5 mg) + IR, and NTC (1 mg) + IR groups received tumor irradiation and either IL-12 GET, or 0.5 mg or 1 mg of B16-F10 NTC, respectively. The main therapeutic groups were the VAC (0.5 mg) + IR and VAC (1 mg) + IR groups, which received tumor irradiation and either 0.5 mg or 1 mg of the B16-F10 vaccine including IL-12 GET, respectively. * = *p* < 0.05 against all other groups unless marked; ns = not significant between groups on each end of the line; ★ = *p* < 0.05 between groups on each end of the line; GET = gene electrotransfer; NTC = 1 unit of non-viable B16-F10 tumor cells (0.5 mg or 1 mg), prepared using the adjusted vaccine preparation protocol; VAC = 1 unit of the B16-F10 vaccine (0.5 mg or 1 mg), prepared using the adjusted vaccine preparation protocol, including IL-12 GET.

**Figure 5 vaccines-08-00111-f005:**
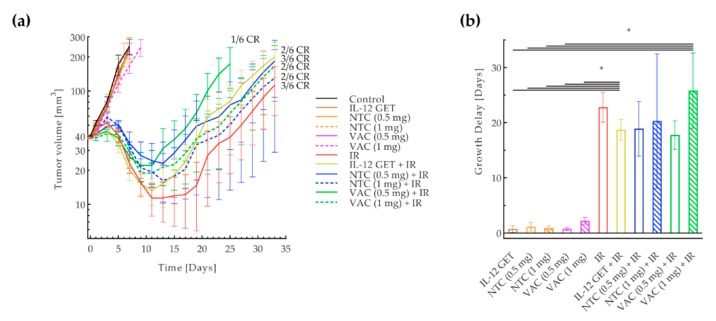
Adjusted CT26 vaccination. (**a**) Tumor growth curve. Tumor volume in a logarithmic scale plotted against time. (**b**) Tumor growth delay based on tumor doubling time. Seven treatment groups with six mice were included in the experiment. The control group received a mock vaccine. The IL-12 GET group received IL-12 GET. The NTC (0.5 mg) and NTC (1 mg) groups received 0.5 mg or 1 mg of CT26 NTC, respectively. The VAC (0.5 mg) and VAC (1 mg) groups received 0.5 mg or 1 mg of the CT26 vaccine including IL-12 GET, respectively. The IR group received tumor irradiation with 15 Gy. The IL-12 GET + IR, NTC (0.5 mg) + IR, and NTC (1 mg) + IR groups received tumor irradiation and either IL-12 GET, or 0.5 mg or 1 mg of CT26 NTC, respectively. The main therapeutic groups were the VAC (0.5 mg) + IR and VAC (1 mg) + IR groups, which received tumor irradiation and either 0.5 mg or 1 mg of the CT26 vaccine including IL-12 GET, respectively. * = *p* < 0.05 between groups on each end of the line; CR = complete response; GET = gene electrotransfer; NTC = 1 unit of non-viable CT26 tumor cells (0.5 mg or 1 mg), prepared using the adjusted vaccine preparation protocol; VAC = 1 unit of the CT26 vaccine (0.5 mg or 1 mg), prepared using the adjusted vaccine preparation protocol, including IL-12 GET.

**Figure 6 vaccines-08-00111-f006:**
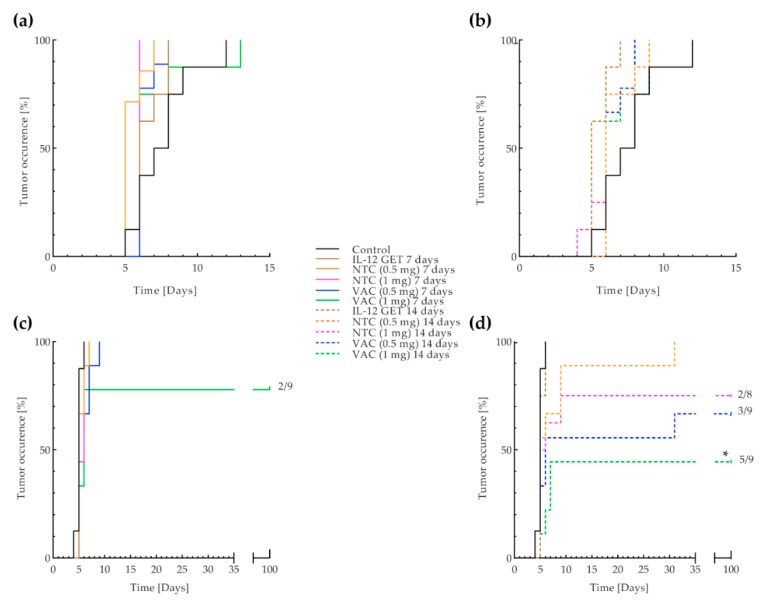
B16-F10 and CT26 preventative vaccination. Kaplan–Meier analysis of (**a**) preventative B16-F10 vaccination, 7 days, (**b**) preventative B16-F10 vaccination, 14 days, (**c**) preventative CT26 vaccination, 7 days, and (**d**) preventative CT26 vaccination, 14 days. Event = tumor occurrence, censored event = no tumor occurrence for 100 days. Eleven treatment groups with 8–9 mice were included in the experiments. The control group received no treatment. The IL-12 GET 7 or 14 days groups received IL-12 GET 7 or 14 days before tumor inoculation. The NTC (0.5 mg) 7 or 14 days groups and NTC (1 mg) 7 or 14 days groups received 0.5 mg or 1 mg of NTC, 7 or 14 days before tumor inoculation, respectively. The main treatment groups were VAC 0.5 mg 7 or 14 days and VAC 1 mg 7 or 14 days, which received 0.5 mg or 1 mg of the B16-F10 or CT26 vaccine, prepared using the adjusted vaccine preparation protocol, including IL-12 GET, 7 or 14 days before tumor inoculation, respectively. * = *p* < 0.05 between the CT26 VAC (1 mg) 14 days group and the Control and IL-12 GET 7 or 14 days groups; GET = gene electrotransfer; NTC = 1 unit of non-viable B16-F10 or CT26 tumor cells (0.5 mg or 1 mg), prepared using the adjusted vaccine preparation protocol; VAC = 1 unit of the B16-F10 or CT26 vaccine (0.5 mg or 1 mg), prepared using the adjusted vaccine preparation protocol, including IL-12 GET.

**Figure 7 vaccines-08-00111-f007:**
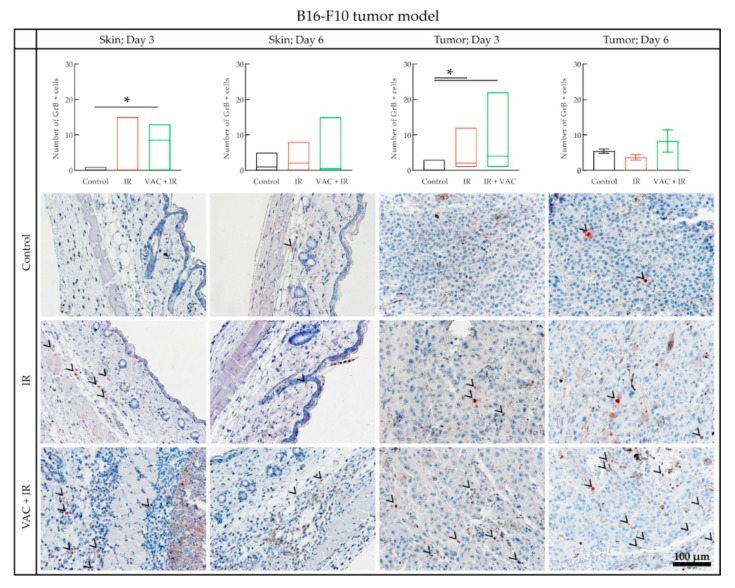
Histological analysis of the B16-F10 tumor model samples. The number of GrB+ cells in IHC stained skin and tumor samples. Data was plotted as the line at the median with the data range (min, max) (Skin; Day 3, Skin; Day 6, Tumor; Day 3) or as the mean ± standard error (SE) (Tumor; Day 6). Representative images of the samples stained for GrB+ cells for each therapeutic group are under the graphs. The black arrowheads indicate the red GrB+ cells. GrB+ = Granzyme B positive cells; * = *p* < 0.05 determined by Dunn’s test for multiple comparison; IHC = immunohistochemically; VAC = 1 unit of the B16-F10 vaccine (0.5 mg), prepared using the adjusted vaccine preparation protocol, including IL-12 GET; IR = tumor irradiation with 15 Gy; GET = gene electrotransfer.

**Figure 8 vaccines-08-00111-f008:**
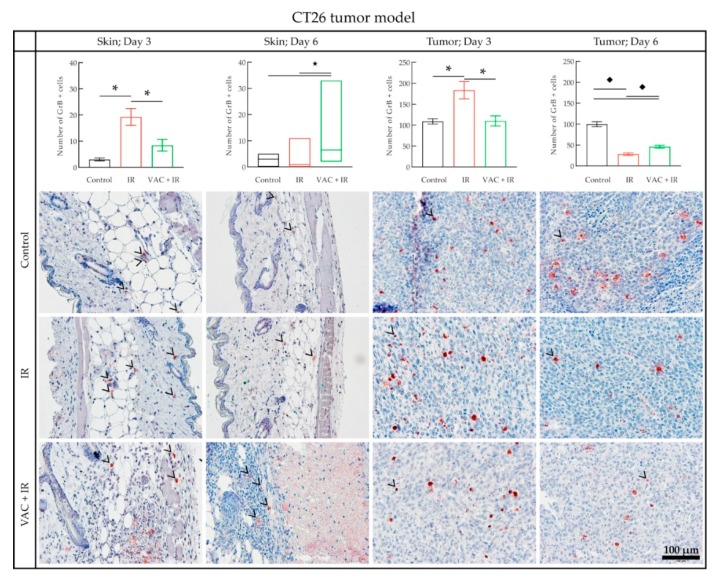
Histological analysis of the CT26 tumor model samples. The number of GrB+ cells in IHC-stained skin samples. Data were plotted as the line at the median with the data range (min, max) (Tumor; Day 3) or as the mean ± standard error (SE) (Skin; Day 3, Skin; Day 6, Tumor; Day 6). Representative images of the samples stained for GrB+ cells for each therapeutic group are under the graphs. The black arrowhead indicates the red GrB+ cells. GrB+ = Granzyme B positive cells; * = P < 0.05 determined by Dunnet’s T3 multiple comparisons test; ★ = *p* < 0.05 determined by Dunn’s multiple comparisons test; ◆ = *p* < 0.05 determined by Tukey’s multiple comparisons test; IHC = immunohistochemically; VAC = 1 unit of the CT26 vaccine (0.5 mg), prepared using the adjusted vaccine preparation protocol, including IL-12 GET; IR = tumor irradiation with 15 Gy; GET = gene electrotransfer.

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
