# Peer review of "Development of Tumor Cell-Based Vaccine with IL-12 Gene Electrotransfer as Adjuvant"

_vaccines, 2020, doi:10.3390/vaccines8010111_

Round 1

Reviewer 1 Report

The manuscript by Remic T et al. describes the antitumor response and immunomudolatory effects using tumor cell-based vaccines combined to in vivo electrotransfer of plasmid DNA encoding IL-12 and irradiation. Optimization of vaccination protocol, investigation of both the therapeutic and preventative effects, and evaluation of local and systemic immune response induced by vaccination in CT26 and B16-F10 tumor models by histological analysis are described. The study provides evidences useful for better understanding of the antitumor potential of tumor cell-based vaccines.

In reviewer opinion, histological analysis of tumor model samples at later time points, as well as of CT26 tumor samples in preventative vaccination setting would help in further characterizing the induced immune response.

Comparison to real world applications should be further discussed.

Author Response

Dear reviewer

Thank you for your time, valuable opinions and suggestions.

The manuscript has been revised to address your comments and the necessary corrections were made to improve the quality of the manuscript, including English editing. The pertinent changes in the revised manuscript are highlighted (yellow - authors’ changes, green – English pre-editing).

Please see the attachment for point-by-point responses.

Reviewer 2 Report

This manuscript described tumor-cell-based vaccines with IL-12 gene electrotransfer using two murine tumor cell lines, B16-F10 and CT26.

The manuscript is basically well written and the experiments are well designed. The results look interesting and promising. The manuscript contains many data with complicated experiments with many different experimental groups. So, it is hard to follow the experiments.

The manuscript should be shorter and more concise. And English should be brushed up.

Why was IL-12 GET performed at the area surrounding the vaccine injection site in pilot vaccination? In adjusted vaccination, IL-12 plasmid was dissolved in NTC suspension solution.

Was the expression of IL-12 at the injection site?

Dis authors confirm that local tumor irradiation induce MHC class I molecule expression?

Figs. 7 and 8. Which data were shown as median with range? And which data were shown as mean ± SE?

Author Response

(The authors gave the same response as above.)

Round 2

Reviewer 2 Report

The manuscript was well improved after revision.

I just comment minor points.

Page 13, lines 359, 360, 394, 405, 406. Is the usage of “hot” tumor and “cold” tumor general? I am not familiar with the usage.

Lines 25, 369, 437, 449, 452, 460. “regime” may be OK, but “regimen” seems better.

Line 55. “ablative” may be OK, but “ablation” seems better.

Line 65. “tumors cells” should be “tumor cells”.

Line 71. The usage of “that” seems strange, Check it.

Line 92. “restriction analysis” reads “restriction enzyme analysis”.

Lines 124, 143. C57Bl/6 reads C57BL/6.

Line 156. “tumor induction” means “tumor inoculation”, doesn’t it?

Author Response

Thank you very much for your valuable time and suggestions.

The manuscript has been revised to address your comments and the necessary corrections were made to improve the quality of the manuscript. The pertinent changes in the revised manuscript are highlighted (blue - authors’ changes).

The point-by-point responses are in the attachment.
